# CRISPR-Cas9 Mediated Stable Expression of Exogenous Proteins in the CHO Cell Line through Site-Specific Integration

**DOI:** 10.3390/ijms242316767

**Published:** 2023-11-26

**Authors:** Zhipeng Huang, Arslan Habib, Guoping Zhao, Xiaoming Ding

**Affiliations:** 1Collaborative Innovation Center for Genetics and Development, State Key Laboratory of Genetic Engineering, Department of Microbiology, School of Life Sciences, Fudan University, Shanghai 200438, China; 2Laboratory of Molecular Immunology, State Key Laboratory of Genetic Engineering, School of Life Sciences, Fudan University, Shanghai 200438, China

**Keywords:** CHO, CRISPR-Cas9, EGFP, HSA, site-specific integration

## Abstract

Chinese hamster ovary (CHO) cells are a popular choice in biopharmaceuticals because of their beneficial traits, including high-density suspension culture, safety, and exogenously produced proteins that closely resemble natural proteins. Nevertheless, a decline in the expression of exogenous proteins is noted as culture time progresses. This is a consequence of foreign gene recombination into chromosomes by random integration. The current investigation employs CRISPR-Cas9 technology to integrate foreign genes into a particular chromosomal location for sustained expression. Results demonstrate the successful integration of enhanced green fluorescent protein (EGFP) and human serum albumin (HSA) near base 434814407 on chromosome NC_048595.1 of CHO-K1 cells. Over 60 successive passages, monoclonal cell lines were produced that consistently expressed all relevant external proteins without discernible variation in expression levels. In conclusion, the CHO-K1 cell locus, NC_048595.1, proves an advantageous locus for stable exogenous protein expression. This study provides a viable approach to establishing a CHO cell line capable of enduring reliable exogenous protein expression.

## 1. Introduction

The CHO cell line stands as the most highly employed cell line in biopharmaceuticals, boasting numerous advantages over alternative options [1]. The CHO cells can be cultured in a serum-free medium with minimal chemical composition. They are exceptionally safe, as they do not serve as a host for human pathogenic viruses. Moreover, the proteins they express after post-translational modification are strikingly similar to human proteins [2]. Nonetheless, a significant concern associated with these cells is the instability of protein expression over extended production periods [3]. The prevalent approach to developing recombinant CHO (rCHO) cell lines for therapeutic protein production relies heavily on random integration and extensive shielding, which constitutes the underlying source of this problem [2]. Additionally, the CHO cell instability stems from chromosome rearrangement, transcriptional inactivation, and genotypic drift during cell culture [4]. This is in agreement with Butler’s findings in 2005, where randomly integrated recombinant CHO cell lines lacked control over gene insertion sites and copy numbers, leading to notable discrepancies between the cell lines [5]. Furthermore, extensive subculture often results in non-producing cells [6], which fail to meet good manufacturing practice (GMP) supervision for drug production [7]. The conventional approach to generating recombinant cell lines requires numerous cycles of shielding and validation for the acquired clones. This integrated process spans from 6 to 12 months, leading to a substantial rise in research and innovation expenses [3]. These aspects constrain the advancement of the CHO cells as a system for pharmaceutical protein expression. However, by using site-directed integration to integrate foreign genes into exact genomic sites, these difficulties can be overcome [8].

Site-specific integration (SSI) stands out as an effective strategy for stabilizing exogenous gene expression. The target gene is positioned in a precise area of the CHO cell genome to achieve this [9]. CRISPR-Cas9 technology, known for its convenience, precision, and efficiency, has seen rapid development [10]. It is extensively utilized in the development of the rCHO cell lines [11]. When developing a rCHO cell line through SSI, initial site selection is a crucial step in the process. There has been published research on the sequencing outcomes of the first, second, and third generations of the CHO cells [12,13,14]. The C12orf35 locus on chromosome 8 in CHO cells, the HPRT locus in CHO cells, the Hipp11 gene in CHO-S cells, and the Kcmf1 gene in CHO-K1 cells serve as examples of potentially viable integration sites [15,16,17]. Other cell lines have also shown stable sites, such as the HPRT and GRIK1 sites in the human fibrosarcoma cell line HT1080 [18,19,20]. Despite these discoveries, durable and eminently expressive SSI sites have not yet been effectively utilized in the production of engineered CHO cell lines. Consequently, it remains crucial to identify and validate sites of stable expression within the CHO cell gene pool. In addition, it remains crucial to develop a methodology for recombinant cell construction based on SSI. In the initial phase of our experiment, we employed the CHO-K1 as the foundational cell line. Through high-throughput sequencing, we identified several loci that demonstrated endurable expression of reporter genes. In our current investigation, we have substantiated the stability of one such expression site, located on chromosome NC_048595.1 of the CHO-K cell, inside an intrinsic region of the Cdk6 gene. To integrate the EGFP reporter gene at this precise location, we employed a CRISPR-Cas9-mediated method. Afterwards, we applied the same method to clone the HSA gene. The work will facilitate the efficient construction of recombinant stable CHO cell lines for expressing exogenous proteins.

## 2. Results

### 2.1. Validation of sgRNA Modification Efficiency

To confirm the successful modification of the constructed sgRNA in the CHO-K1 cells, we introduced a targeted plasmid containing the Cas9 protein. Gene editing would require DNA homologous directed repair at the position of mismatched openings. The sgRNA plasmid, CD513B-Cas9 plasmid, and donor plasmid were transfected into CHO-K1 cells using Lipofectamine^TM^ 3000. After 48 h of culture, the cells were selected under 10 µg/mL puromycin pressure. Following the death of all cells in the control group, surviving cells were collected for further procedures. The monoclonal cells expressing only green fluorescence (GF) were isolated using flow cytometry and seeded individually into 96-well plates. After 6–7 days of cultivation, the cells from wells containing only one cell cluster were selected for further development. The genomic DNA was then purified and verified via PCR. The SSI of the target gene was identified using 5′ and 3′ junction PCR. To assess whether the monoclonal cells were homozygous or heterozygous, an out-to-out PCR was conducted. Table 1 summarizes the primer sequences used in the expression system. A schematic diagram of the site-specific integration is shown in Figure 1 and Figure 2 [21].

Subsequently, we conducted a PCR amplification of the sequences both upstream and downstream of the Cas9 cleavage site in the transfected cell pool, followed by digestion using T7 endonuclease I. Table 2 displays the primer sequences. The amplified PCR fragment measured 865 base pairs in size. As T7 endonuclease I can identify and cleave mismatched DNA double strands, a gene editing event would manifest as two distinct bands, one at 523 base pairs and the other at 342 base pairs, visible on agarose gel electrophoresis following enzyme digestion, as illustrated in Figure 3a. After the PCR, the products were subjected to sequencing. The findings, as depicted in Figure 3b, revealed a pronounced peak nesting just after the fourth base upstream of the PAM sequence. The region between three and four bases upstream of the PAM sequence was identified by the Cas9 complex. These findings affirm that the constructed sgRNA effectively guided the Cas9 protein to the specific location within the CHO-K1 genome.

### 2.2. Selection and Characterization of Monoclonal Cell Lines with Integrated EGFP Gene

After verifying the effective editing of the target site by the sgRNA plasmid, the 600-bp DNA sequences upstream and downstream of the PAM site in the BB sequence were selected as the 5′ and 3′ homologous arms, respectively. Between these arms, an EGFP donor plasmid was designed, containing a green fluorescent gene expression (GFGE) cassette (hPGK-EGFP-SV40 polyA) and a puromycin gene expression cassette (EF-1α -PuroR-SV40 polyA). Additionally, a red fluorescent gene expression cassette (CMV-mCherry-SV40 polyA) was positioned upstream of the 5′ homologous arm.

In accordance with the devised SSI approach, cell lines showing successful SSI events demonstrated expression of the EGFP gene, manifesting solely as green fluorescence (GF). A red fluorescent gene expression cassette, mCherry, was constructed upstream of the 5′ homologous arm. Subsequently, the plasmid containing mCherry was transfected into CHO-K1 cells. Conversely, cell lines with random integration events exhibited co-expression of both the EGFP and mCherry genes. Cell lines lacking integration events displayed no detectable fluorescence. Subsequently, the transfected cells underwent collection and sorting through flow cytometry, as depicted in Figure 4A. When employing the K1 cells as the negative control, it was determined that 21% of the cells exclusively emitted green fluorescence, indicating successful site-directed integration events. These monochromatic, green-emitting cells were then isolated into individual wells of 96-well plates, with each well containing a single cell.

Genomic material from the monoclonal cell lines was then isolated for the PCR analysis, as shown in Figure 4B,C. The PCR stock for the 5′-homologous arms measured approximately 1830 bp, while those for the 3′-homologous arms measured about 2500 bp, aligning with our anticipated results. Through 5′ and 3′ junction PCR, we identified four positive monoclonal strains (K1-EGFP-8, K1-EGFP-15, K1-EGFP-21, K1-EGFP-24) out of the 32 tested. The results of the out-out PCR (Figure 4D) revealed that K1-EGFP-8, K1-EGFP-21, and K1-EGFP-24 displayed both full knock-in 3681-bp bands and 1200-bp bands without knock-in, indicating their heterozygous nature. Conversely, the K1-EGFP-15 only exhibited 1200-bp bands without a knock-in. Sequencing corroborated that the 3681-bp band contained the site-integrated complete EGFP gene, while the 1200-bp band contained unintegrated chromosomal DNA, aligning with our anticipated results. Positive monoclonal cell lines displayed GF in flow cytometry investigation. These findings affirm that the CRISPR-Cas9 approach can effectively facilitate the integration of foreign genes into this site, resulting in stable expression.

### 2.3. Stability of EGFP Expression over 60 Passages in Site-Specific Integrant Cell Lines

The CHO cell lines harbouring the EGFP reporter gene underwent PCR screening over 60 consecutive culture generations. Each cell line was initiated at a density of 1 × 10^6^ cells/mL in 50 mL of CD CHO media, supplemented with L-glutamine at a concentration of 8 mmol/L. Three concurrent tests were executed. Throughout the cell passages, measurements for cell viability, cell density, and average fluorescence intensity were recorded every 10 generations. The cell lines K1-EGFP-8, K1-EGFP-21, and K1-EGFP-24, which were successfully adapted to suspension culture, underwent continuous passage for 60 generations. Flow cytometry was employed to assess the EGFP gene expression every 10 generations. As depicted in Figure 5, all the positive monoclonal strains maintained stable EGFP gene expression in suspension culture throughout each generation. The average fluorescence intensity hovered around 11,000 RFU. Although there was a slight fluctuation in the average fluorescence intensity throughout the passage, the variation never exceeded 15,000 RFU. This observation suggests that the integration site remains stable.

### 2.4. Selection and Characterization of Monoclonal Cell Lines with Integrated HSA Gene

Upon the demise of all cells in the control group, the surviving cells from the experimental group were gathered, expanded in culture, and subjected to flow cytometry analysis. The cells exhibiting exclusive green fluorescence without concurrent red fluorescence were isolated, as depicted in Figure 6A. A noteworthy 37.2% of the cells demonstrated primarily green fluorescence, implying potential site-specific integration events within this subset. These cells were then selected for further culture expansion.

The PCR assays were employed to corroborate the integration of exogenous genes, while Western blot analysis was utilized to assess the HSA expression in the culture supernatant. The electrophoretic patterns of PCR stock (Figure 6B,C) revealed product sizes of approximately 4100-bp for the 5′-homologous arms and 5000-bp for the 3′-homologous arms. Through 5′ junction and 3′ junction PCR, 12 monoclonal strains K1-HSA-5, 9, 18, 24, 26, 30, 36, 40, 41, 44, 46, and 54 exhibited bands of the correct size. Results from out–out PCR (Figure 6D) confirmed that K1-HSA-5, K1-HSA-9, K1-HSA-18, K1-HSA-24, K1-HSA-26, K1-HSA-40, K1-HSA-41, K1-HSA-44, K1-HSA-46, and K1-HSA-54 all displayed both full knock-in 6000-bp bands and 1200-bp bands without knock-in, indicating the procured positive monoclonal cells were heterozygotes. This outcome aligned precisely with our expectations. Subsequently, K1-HSA-18, K1-HSA-40, and K1-HSA-46 monoclonal cell lines were determined for further expansion based on band concentration and brightness. These findings affirm the efficacy of the aforementioned methods in achieving SSI and the subsequent HSA expression at this targeted locus.

### 2.5. HSA Expression in Site-Specific Integrant Cell Lines over 60 Passages

The Western blot analysis demonstrated that the monoclonal strains K1-HSA-18, K1-HSA-40, and K1-HSA-46 exhibited robust HSA gene expression (Figure 7). Following suspension adaptation, these three positive monoclonal strains underwent assessments for the HSA gene expression stability and yield through batch and flow culture methods. In Figure 8, the batch culture of the K1-HSA-18, K1-HSA-40, and K1-HSA-46 positive monoclonal strains is depicted over 60 generations at intervals of 0, 10, 20, 30, 40, 50, and 60 generations. The figure illustrates a progressive improvement in K1-HSA-18 growth, which is correlated with a gradual increase in the HSA expression levels, indicative of enhanced cellular performance. At the 60th generation, the maximum cell density reached 8.53 × 10^6^ cells/mL, accompanied by an HSA expression level of 16.2 mg/L.

## 3. Discussion

The CRISPR/Cas9 is a potent tool for manipulating the CHO cells to meet biopharmaceutical research requirements. Although it has previously been discussed the topic of SSI of foreign genes into the CHO cells using CRISPR/Cas9, no pharmaceutical utilization was demonstrated [6]. In this study, our objective is to leverage CRISPR/Cas9-mediated SSI to identify potentially highly transcriptionally active sites within the CHO cells. This pursuit aims to enhance the advancement of stable cell lines for application in industrial settings.

The current study introduces a methodology employing the CRISPR-Cas9 strategy. Specifically, the EGFP and HSA genes were successfully integrated into the CHO-K1 cell chromosome NC_048595.1, positioned proximal to the 434,814,407th base, precisely within the intron region of the Cdk6 gene. Throughout 60 passages, the cells underwent subculturing, and the expression of exogenous proteins was evaluated at 60-passage intervals. The findings demonstrated that all monoclonal cell lines expressed their respective foreign proteins. This leads to the conclusion that the genomic region encompassing the 434,814,407th base of chromosome NC_048595.1 functions as a reliable integration site for foreign protein expression. Our initial step involved demonstrating the viability of this approach by incorporating HSA into the NC_048595.1 within the CHO-K1 cells. Subsequently, we scrutinized three potential chromosomal locations (A, B, C) for targeted integration of two prototypical model proteins, EGFP and HSA, in the CHO-K1 cells. To refine the approach, we employed two plasmids for transfecting the CHO cells: one containing the sgRNA and the other serving as the donor plasmid. These sgRNAs were classified into three types based on their targeting sites. This method significantly increased the targeting efficiency, elevating it from the previously reported range of 10~30% to an impressive 72%. This enhancement also ensured the precise integration of exogenous genes at designated sites [22]. Stable cell line development is based on productivity and stability. Through extensive multi-passage cultivation, it was evident that the stable cell lines integrated at NC_048595.1 exhibited superior productivity and stability for both EGFP and HSA proteins. The EGFP and HSA clones at NC_048595.1 demonstrated an 80-fold increase in expression levels compared to randomly integrated clones. Furthermore, the levels of the EGFP and HSA proteins in CHO-K1 cells remained consistent over 60 passages. These findings underscore that the introduction of exogenous genes into the NC_048595.1 locus can maintain production levels and cell stability for 60 passages. This provides robust evidence for the NC_048595.1 locus’ status as a transcriptional hotspot in the CHO cell lines. In a comparable study conducted by Ding et al. (2023), it was demonstrated that the exogenous proteins EGFP and HSA were effectively incorporated near the base 1969647 on chromosome NW_003613638.1 of CHO-K1 cells. The resultant monoclonal cell lines, established through 60 consecutive passages, consistently manifested the expression of all associated exogenous proteins, with no notable variations in expression levels detected [23]. Zhao M et al. (2018) employed a similar site-specific integration approach, exploring three sites C12orf35, HPRT, and GRIK1 with potential high transcriptional activities in CHO cells. The objective was to identify potential transcriptional hotspots and establish a robust strategy for site-specific integration, facilitating the efficient development of recombinant cell lines. The results indicated the viability of C12orf35 as the preferred target site for exogenous gene integration. The findings strongly support the notion that CRISPR/Cas9-mediated targeted integration at C12orf35 is a dependable strategy for the swift generation of the recombinant CHO cell lines [8].

One limitation of this study pertains to the productivity data, which was obtained from shake flask cell cultures rather than from bioreactors. Assessing the expression of recombinant proteins in a bioreactor setting would offer a more industry-relevant perspective and warrants further investigation. It is worth noting that the conventional random integration strategy typically necessitates multiple rounds of pressure selection. This extended the timeline for constructing high-yielding rCHO cell lines to approximately 6–12 months [24]. The utilization of the site-specific integration strategy outlined in this study presents a significant advantage in terms of time and cost savings. This approach requires only 2 weeks to obtain targeted transfected pools and 4 weeks to select integrated clones. Moreover, it is crucial to highlight that in accordance with the FDA’s stipulations, non-productive cell clones should not surpass 30% during the scale-up and production process, extending from the master cell bank to the working cell bank [4].

Hence, the cell strains developed through SSI offer a more streamlined path to meeting this criterion. This aligns more closely with good manufacturing practice standards in pharmaceutical production. Given the relatively low cell density (3 × 10^6^/mL) and less than optimal environment in shake flasks, there exists promising potential to enhance productivity by a factor of 5 to 10 through the refinement of cell culture conditions. This may encompass adjustments in media composition, supplementation, dissolved oxygen levels, inclusion of trace elements, or even the implementation of a perfusion-fed batch process [25]. Furthermore, strategies rooted in cell biology can enhance the production of target proteins, potentially by targeting key molecules involved in cell cycle regulation, apoptosis, or metabolism [26].

The primary challenge facing this site-specific integration method is the occurrence of off-target effects induced by the CRISPR/Cas9. Studies have indicated that mismatches at the base level in the sgRNA may be the primary factor contributing to off-target mutations [27]. The target specificity of each sgRNA editing sequence has previously been assessed by identifying possible off-target locations [6]. In both research and manufacturing, improving efficiency and stability in cell line generation has been a challenge for many years. The current experiment successfully pinpointed an optimal integration site, NC_048595.1, and devised a reliable SSI approach facilitated by CRISPR/Cas9. This led to the establishment of stable cell lines demonstrating high productivity and remarkable stability. Furthermore, other noteworthy techniques have been developed for the development of recombinant cell lines, such as the AGIS system, which enables the enhancement of target protein productivity [28]. Another noteworthy technique is the CRIS-PITCH system, which involves considerably shorter and simpler microhomology sequences for targeting [29]. The generation of cell lines has enormous potential with the use of these cutting-edge techniques characterized by high productivity, stability, and efficiency, catering specifically to the demands of the biopharmaceutical industry.

## 4. Materials and Methods

### 4.1. Cell Culture

The adherent cells were cultured in Ham’s F-12 K medium with 10% FBS at 37 °C in a 5% CO_2_ incubator [22] (Sun 2015). Suspension cells were cultivated in a shaker using CD-CHO medium, supplemented with 8 mM L-glutamax, at 110 rpm, in the humified environment at 37 °C with 5% CO_2_.

### 4.2. Expression System Construction and Transfection

Based on the previous high-throughput sequencing results in the laboratory, three sequences, namely BB (5′-TATCTTTGCAAATACAGTGA-3′), VI (5′-TGCAACTCTCAGATCTAACT-3′), and HW (5′-ACCCTTGTGCCCCAAAGACA-3′), which had the highest read counts, were chosen as the target sequences. The primers sgRNA-F and sgRNA-R were annealed in the following systems: sgRNA-F (4 µL), sgRNA-R (4 µL), NEBuffer2 (2 µL), and ddH_2_O (10 µL), in a water bath at 95 °C for 5 min. The constructed sgRNA plasmid was transfected into CHO-K1 cells using Lipofectamine^TM^ 3000 reagent at a ratio of 1.8:1. After 72 h, the cells were harvested, and genomic DNA was purified and amplified via PCR via primers BB-F, BB-R, VI-F, VI-R, HP-F, and HP-R. Subsequently, the PCR stock underwent digestion with T7 endonuclease I.

### 4.3. Donor Plasmid Design and Recombinant Monoclonal Confirmation

The HSA donor plasmid is based on the EGFP donor plasmid with the inclusion of a HAS expression cassette (CMV-HSA-SV40 polyA) inserted between the 5′ homologous arm and the (GFGE) cassette (Figure 2). When the majority of cells in the 24-well plate reached over 90%, DNA was purified through a mini-genomic DNA extraction kit (Beyotime, Shanghai, China) from a monoclonal cell line for PCR amplification. The cell digestion solution was transferred into an Eppendorf tube, the supernatant was centrifuged, and then 200 µL of PBS was used for washing and another round of centrifugation. Afterwards, 100 µL of 1% Triton solution dissolved in lysis buffer and 10 µL of protease K mg/mL (Magen Biotechnology Co., Ltd. Guangzhou, China) were added, followed by incubation in a water bath at 56 °C for 1 h and then a water bath at 95 °C for 10 min. The Eppendorf tube lid should be carefully opened and closed promptly to prevent evaporation of the solution which could affect extraction efficiency. Finally, the sample was centrifuged at 13,000 rpm for 15 min, and the supernatant was collected, and stored in the freezer at −30 °C.

The sgRNA plasmid, CD513B-Cas9 plasmid, and donor plasmid were transfected into CHO-K1 cells using Lipofectamine^TM^ 3000 (Thermo Fisher Scientific, Shanghai, China). After 48 h of culture, the cells were selected under 10 µg/mL puromycin pressure. Following the death of all cells in the control group, surviving cells were collected for further procedures. The monoclonal cells expressing only green fluorescence (GF) were isolated using flow cytometry and seeded individually into 96-well plates. After 6–7 days of cultivation, cells from wells containing only one cell cluster were selected for further development. The genomic DNA was then purified and verified via PCR. SSI of the target gene was identified using 5′ and 3′ junction PCR. To assess whether the monoclonal cells were homozygous or heterozygous, an out-to-out PCR was conducted. Table 2 displays the primer sequences.

### 4.4. Determination of Targeted Protein Expression

EGFP gene expression in SSI cell lines was evaluated through flow cytometry. A BD FACS AriaIII flow sorter was used to determine the average GF intensity of 20,000 cells using CHO-K1 blank cells as a negative reference. The outcomes were observed by FlowJo V10 software.

For the detection of the HSA gene expression in cell lines with SSI, the Western blot analysis was conducted as follows: 20 µL of denatured cell culture supernatant was loaded onto a 10% SDS-PAGE gel. The proteins were moved from the gel to an NC membrane for 70 min at 110 V. Subsequently, the NC membrane was air-dried and blocked for one hour at room temperature with 5% skim milk. After washing, the membrane was incubated at 4 °C overnight with an HSA antibody (ab10241) from Abcam in Shanghai, China. The membrane was then exposed two more times with TBST before being treated with a goat anti-mouse antibody for 1.5 h at room temperature. Finally, the TBST on the NC film was drained, and it was placed in the gel imager. The film was evenly coated with 200 µL of ECL hypersensitive developer and developed.

### 4.5. Identification of Cell Line and Exogenous Protein Stability

Recombinant cell lines are assessed for subculture and cryopreservation stability. Subculture stability pertains to product expression constancy during successive rounds of cell subculture. In contrast, cryopreservation stability involves evaluating product expression after periodic resuscitation of cells previously cryopreserved in liquid nitrogen [21] (Laura 2012). The CHO cell lines containing the reporter gene EGFP, or antibody protein HSA were subjected to the PCR screening for 60 consecutive culture generations. Each cell line was seeded at a density of 1 × 10^6^ cells/mL and 50 mL of the CD CHO media supplemented with L-glutamine at a concentration of 8 mmol/L. Three simultaneous tests were conducted. Throughout the cell passage, measurements were taken every 10 generations for cell viability, cell density, average fluorescence intensity, and HSA protein content. Cell line alterations were assessed via flow cytometry, while the average protein content was determined through western blot analysis, serving as an evaluation indicator. The trial was concluded once cell growth fell below 80%.

## 5. Conclusions

This study successfully demonstrated the stability of exogenous protein expression in a specific genomic site located within the region spanning 434,813,807–434,815,007 bp on the chromosome NC_048595.1. An effective approach utilizing CRISPR–Cas9 technology was developed, enabling the successful integration of both the EGFP and HSA genes into this specific site on the CHO-K1 cell chromosome. Monoclonal cell lines were systematically selected via flow cytometry and effectively adapted to suspension culture. These cells were subcultured over 60 passages, with the exogenous protein expression (EGFP and HSA) evaluated every 10 passages. The results showed that there was consistent expression of the respective foreign proteins across all monoclonal cell lines. Hence, it can be concluded that the genomic region proximal to the 434,813,807–434,815,007 bp locus on the chromosome NC_048595.1 in CHO-K1 cells serves as a stable integration site for reliable foreign protein expression.

## Figures and Tables

**Figure 1 ijms-24-16767-f001:**
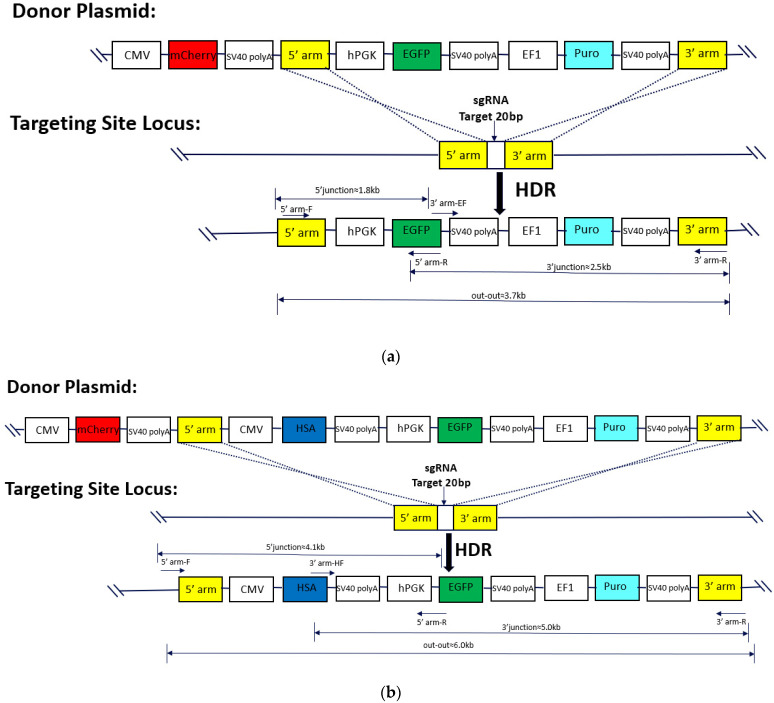
Site-specific integration of foreign genes into CHO-K1 genome. (**a**) Schematic diagram of EGFP gene site-specific integration into CHO-K1 genome; (**b**) Schematic of HSA gene site-specific integration into CHO-K1 genome.

**Figure 2 ijms-24-16767-f002:**
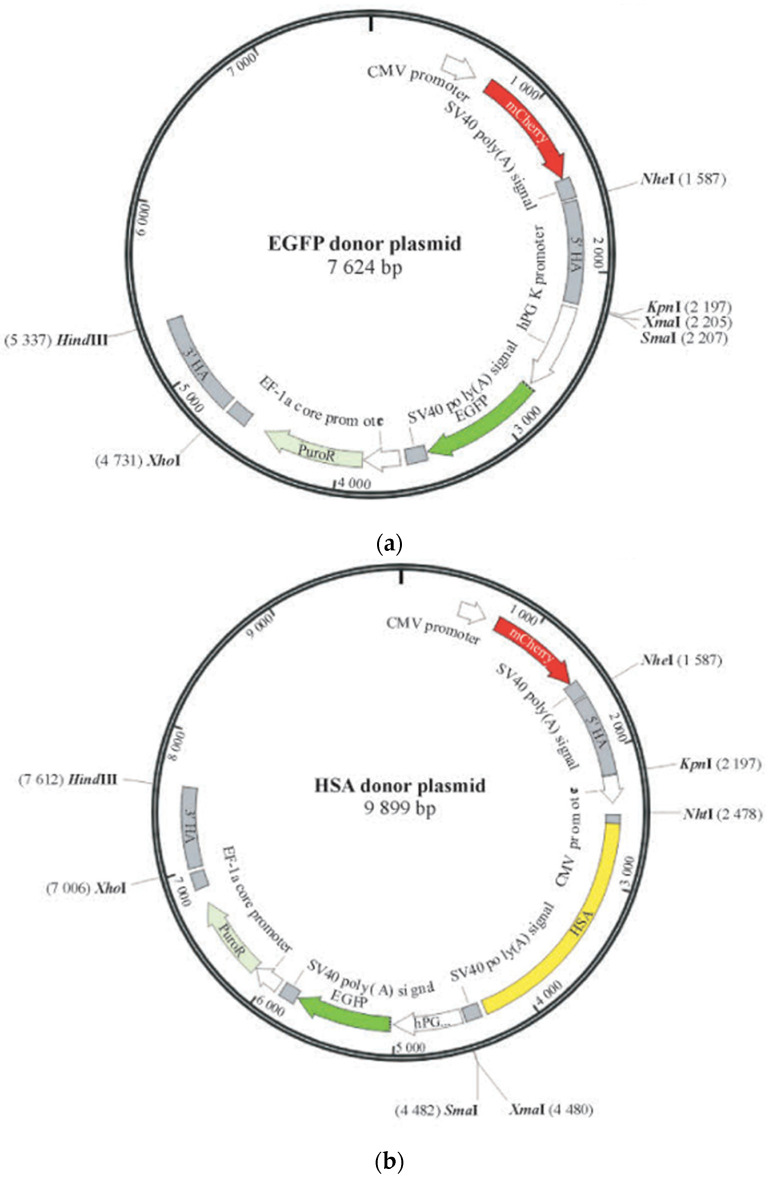
Map of donor plasmids. (**a**) Map of the EGFP donor plasmid; (**b**) Map of the HSA donor plasmid.

**Figure 3 ijms-24-16767-f003:**
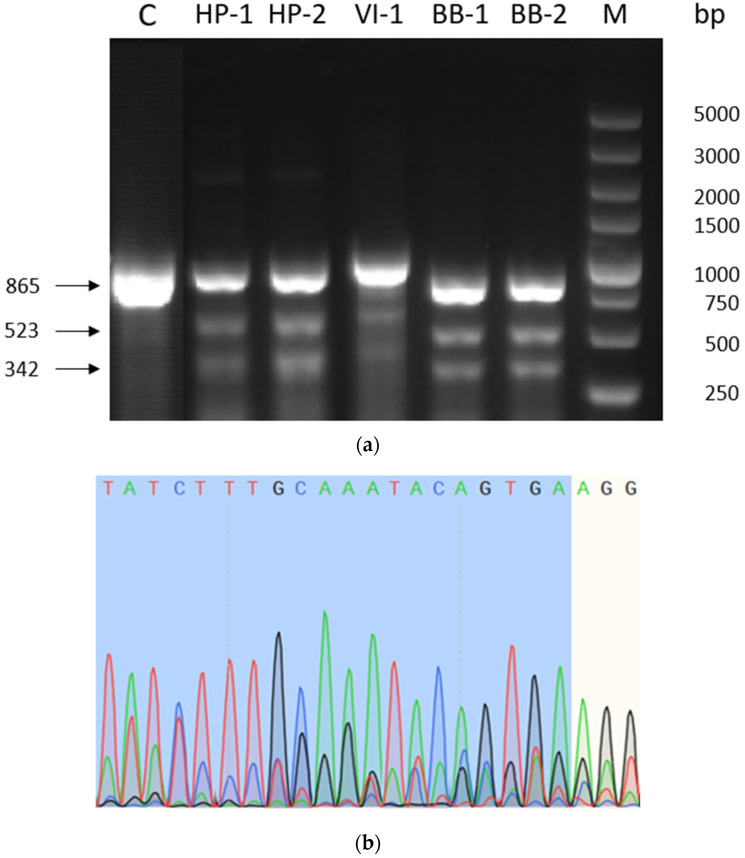
Results of the sgRNA editing efficiency verification. (**a**) Agarose gel electrophoresis of T7E1 digests of PCR products of the cell pool genome transfected with the plasmids sgRNA-Cas9, M: DL5000 marker, lanes 1–6 are: CHO-K1 not transfected with the plasmids sgRNA -Cas9, CHO-K1 transfected with the plasmids sgRNA-Cas9 HP-1, HP-2, VI-1, BB-1 and BB-2; (**b**) Sequencing of genomic PCR products of K1 cells transfected with sgRNA-Cas9 plasmids. Different colored alphabetic orders and peaks represent the nucleotide sequence after sequencing results.

**Figure 4 ijms-24-16767-f004:**
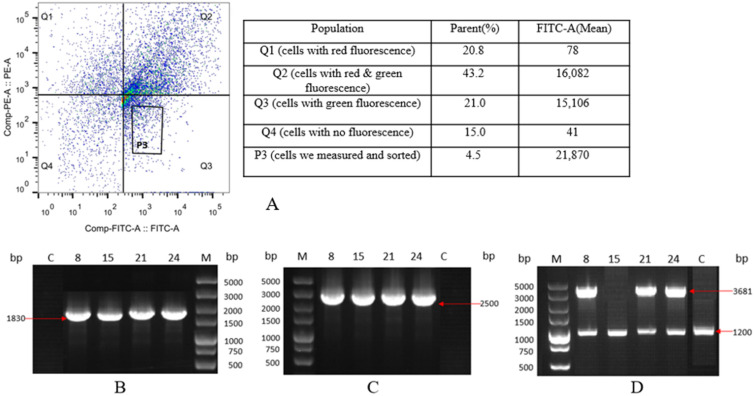
CHO-K1 cell sorting and identification results of site-specific integration of the EGFP gene. (**A**) Flow cytometric sorting of CHO-K1 cells with the EGFP gene introduced by site-directed integration; (**B**) Results of 5′ junction PCR, M: DL 5000 DNA marker, lanes 1–5 are: control, K1-EGFP-8, K1-EGFP-15, K1-EGFP-21, K1-EGFP-24; (**C**) Results of 3′ junction PCR, M: DL 5000 DNA marker, lanes 1–5 are: K1-EGFP-8, K1-EGFP-15, K1-EGFP-21, K1-EGFP-24, control; (**D**) Results of out–out PCR, M: DL 5000 DNA marker, lanes 1–5 are: K1-EGFP-8, K1-EGFP-15, K1-EGFP-21, K1-EGFP-24, control.

**Figure 5 ijms-24-16767-f005:**
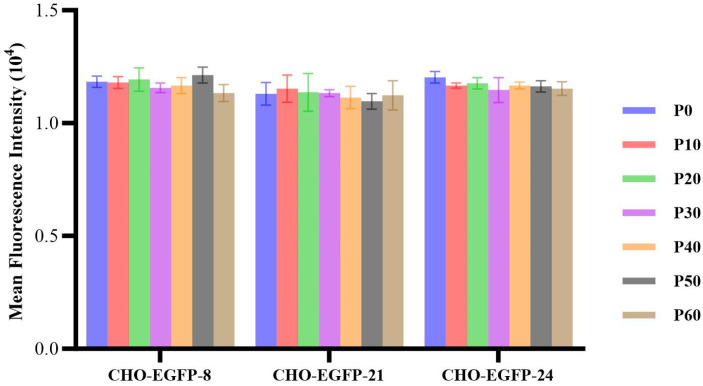
Results of the average fluorescence intensity of three CHO-K1-EGFP suspension cell lines. P0-P60 is the passage of EGFP clones.

**Figure 6 ijms-24-16767-f006:**
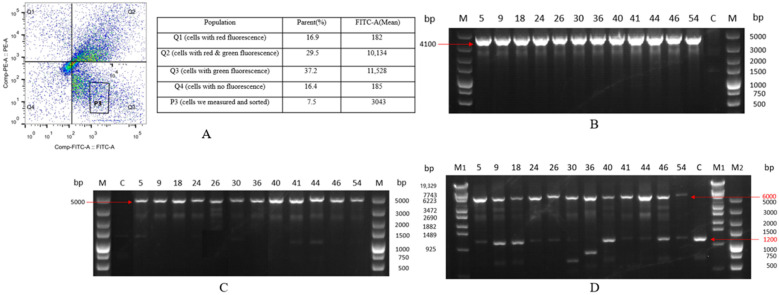
CHO-K1 cell sorting and identification results of site-specific integration of the HSA gene. (**A**) Flow cytometric sorting of CHO-K1 cells with the HSA gene introduced by site-directed integration; (**B**) Results of 5′ junction PCR, M: DL 5000 DNA marker, lanes 1–13 are: K1-HSA-5, K1-HSA-9, K1-HSA-18, K1-HSA-24, K1-HSA-26, K1-HSA-30, K1-HSA-36, K1-HSA-40, K1-HSA-41, K1-HSA-44, K1-HSA-46, K1-HSA-54, control; (**C**) Results of 3′ junction PCR, M: DL 5000 DNA marker, lanes 1–13 are: control, K1-HSA-5, K1-HSA-9, K1-HSA-18, K1-HSA-24, K1-HSA-26, K1-HSA-30, K1-HSA-36, K1-HSA-40, K1-HSA-41, K1-HSA-44, K1-HSA-46, K1-HSA-54; (**D**) Results of out-out PCR, M1: λ-EcoT14 I DNA Marker, M2: DL5000 DNA marker, lanes 1–13 are: K1-HSA-5, K1-HSA-9, K1-HSA-18, K1-HSA-24, K1-HSA-26, K1-HSA-30, K1-HSA-36, K1-HSA-40, K1-HSA-41, K1-HSA-44, K1-HSA-46, K1-HSA-54, control.

**Figure 7 ijms-24-16767-f007:**
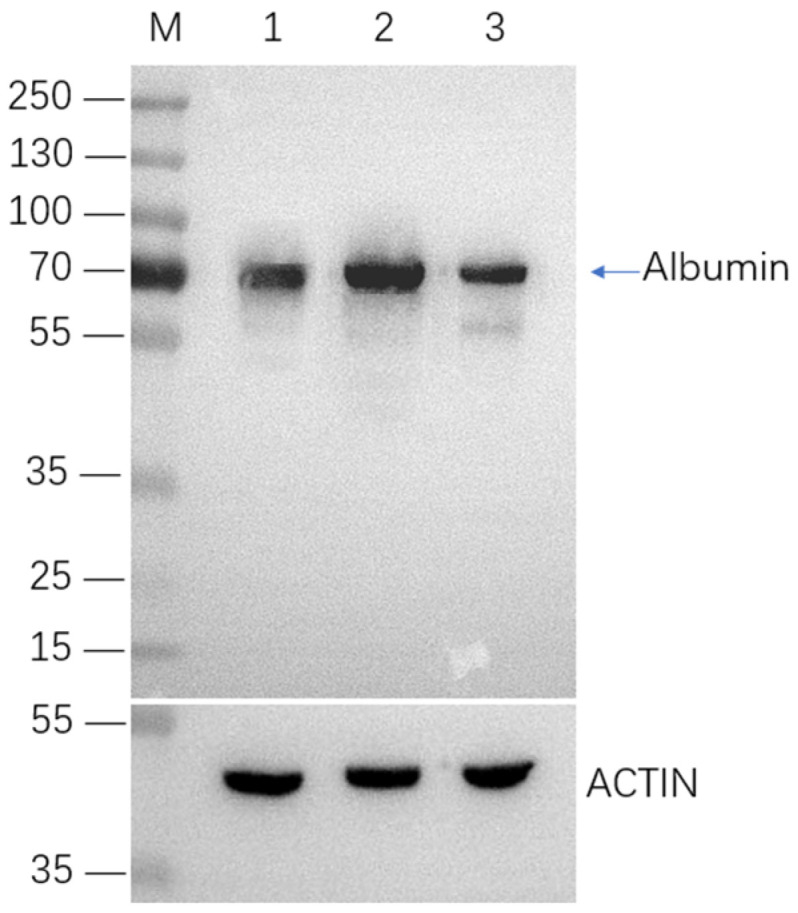
Western blot assay results of HSA gene expression in CHO-K1 cells. M: molecular mass marker, lanes 1–3 are: K1-HSA-40, K1-HSA-18, K1-HSA-46.

**Figure 8 ijms-24-16767-f008:**
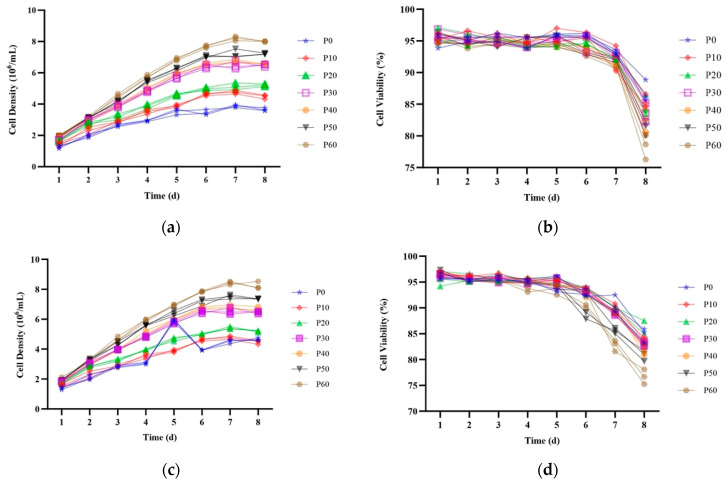
Identification of the HSA gene expression and batch culture of CHO-K1-HSA suspension cell lines. (**a**) K1-HSA-18 cell density changes; (**b**) cell viability changes during batch culture; (**c**) K1-HSA-40 cell density changes; (**d**) HSA cell viability changes during batch culture; (**e**) K1-HSA-46 cell density changes; (**f**) HSA cell viability changes during batch culture; (**g**) Quantitative analysis of HSA protein expression during cell culture.

**Table 1 ijms-24-16767-t001:** Primer’s sequence related to sgRNA.

Primer	Primer Sequence (5′→3′)
sgRNA-F	AGCCTAGTGCTCCTGATACG
sgRNA-R	AGACCGATACCAGGATCTTG
BB-F	AGCCTAGTGCTCCTGATACG
BB-R	TTCTGCTGTGGACTCTGAAG
VI-F	AACCACCAGGTCAGAAATCC
VI-R	CAGAGGCCAATCAGCAGTAG
HP-F	GCTGTGCATTGAAACCCATG
HP-R	TGGGTATGGAGATGGGGCGG

**Table 2 ijms-24-16767-t002:** Primers used for PCR.

Primer	Primer Sequence (5′→3′)
5′ arm-F/Oo-F	CCTGTCCACGTCTAAGTATC
5′ arm-R	GTCCATGCCGAGAGTGATCC
3′ arm-EGFP-F	AAGGGCGAGGAGCTGTTCAC
3′ arm-HSA-F	GGTACTGCTGCTCTGGGTTC
3′ arm-R/Oo-R	GCTTTGCTCCGAAGTCCATC

## Data Availability

Data is contained within the article.

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
