# Peer review of "CRISPR-Cas9 Mediated Stable Expression of Exogenous Proteins in the CHO Cell Line through Site-Specific Integration"

_ijms, 2023, doi:10.3390/ijms242316767_

Round 1
Reviewer 1 Report
Comments and Suggestions for Authors
Please see attached pdf

Written expression is generally ok
Author Response
Dear Editor
International Journal of Molecular Sciences
SUBJECT: ARTICLE RE-SUBMISSION (REVISED)
The subjected paper entitled “CRISPR-Cas9 mediated stable expression of exogenous proteins in the CHO cell line through site-specific integration” is revised according to the referee’s suggestions. The revision is now up to the mark as advised by the potential reviewers. The revised version of our paper has been submitted for publication.
We hereby present you with a list of all the changes/corrections we made in our revised manuscript, along with the Track Changes feature applied in the main file:
Reviewer-1
- In particular, their green/red selec9on scheme is clearly based on Hamaker and Lee (2020) Biotech J. and this paper should be cited.
Thank you for your valuable suggestions. Following your instructions, we have modified the manuscript and included the reference to Hamaker (2020).
- The main problem is that many of the details in the Methods sec9on should be included in the Results – otherwise it is impossible to make sense of the Results. For example, the Results sec9on begins “To confirm the successful modifica4on of the constructed sgRNA in CHO-K1 cells...”. What sgRNA? What Cas9 plasmid was used? How was it transfected? Then Sec9on 2.2 begins “In accordance with the devised SSI approach...” What devised SSI approach? The approach needs to be explained properly before the reader can make sense of it. Too many of these details are included in the Methods sec9on, but not present in Results. The manuscript needs to be restructured so that the Results sec9on can be read as a stand-alone document, with reference made to the Methods sec9on only when specific details of the methods are required. The figure order runs 4,3,1,2 so clearly the plasmid and HDR schemes (which really do need to be in Results) have become misplaced.
Thank you for your suggestions. We have revised the manuscript to enhance the precision of the results description for readers. Additionally, we have adjusted the order of the figures to ensure accuracy in formatting.
I would like to provide a brief overview of sgRNA and its transfection. The sgRNA plasmid is utilized to introduce the target gene sequence into the specified cell genome for gene editing, encompassing processes such as gene knock-in and gene knock-out. The genes encoding the Cas9 protein and sgRNA are introduced into a cell, programmed to alter its target gene, with the Cas9 protein playing a crucial role in recognition. The constructed sgRNA plasmid was transfected into CHO-K1 cells using Lipofectamine™ 3000 reagent at a ratio of 1.8:1. After 72 hours, the cells were harvested, and genomic DNA was purified and amplified via PCR using primers BB-F, BB-R, VI-F, VI-R, HP-F, and HP-R. Subsequently, the PCR stock underwent digestion with T7 endonuclease I.
- As the authors identify, CRISPR has been used for site specific recombination in CHO cells for several years, as described in numerous papers. The search continues for appropriate genome sites that result in high, stable expression of the resultant transgenes. The integration work and isolation of the expressing transgenes is not novel, and the demonstrated stability of expression is encouraging but not wholly convincing as the maximum number of 60 generations is relatively short.
The number of generations required for stable expression of recombinant proteins in CHO cell lines using CRISPR-Cas9 can vary depending on several factors. Achieving stable and high-level expression of the desired protein involves optimizing multiple parameters. In our study, the expression level remained stable for 60 generations, a benchmark supported by previous studies, as referenced here.
Furthermore, through extensive multi-passage cultivation, it became evident that stable cell lines integrated at NC_048595.1 exhibited superior productivity and stability for both EGFP and HSA proteins. EGFP and HSA clones at NC_048595.1 demonstrated an 80-fold increase in expression levels compared to randomly integrated clones. Moreover, the levels of EGFP and HSA proteins in CHO-K1 cells remained consistent over 60 passages. However, additional studies are needed for a deeper understanding of site-specific integration and stability across different generations over time.
Dahodwala H, Lee KH. The fickle CHO: a review of the causes, implications, and potential alleviation of the CHO cell line instability problem. Curr Opin Biotechnol. 2019; 60:128–137.
Schiff LJ. Review: Production, characterization, and testing of banked mammalian cell substrates used to produce biological products. In Vitro Cell Dev Biol Anim. 2005;41(3):65–70.
- Unfortunately, the most interesting aspect of this work is how the authors selected the Cdk6 gene as an integration site. It is unclear whether this was the authors work or due to others, because no details are given about site selection, and no other research papers are cited to jus9fy the use of this site. The authors say at line 227 that they “scrutinized three chromosomal locations (A,B,C) for targeted integrati..” but they do not identify A, B or C, nor to the describe the basis on which they selected these sites. At line 64 they claim “we identified several loci that demonstrated endurable expression of reporter genes” but this not work is neither explained nor cited. At the very least a brief description of this work is required. At line 288 they say “Based on the previous high throughput sequencing results...” but no details or citations are given.
Thank you for the comment. In the present study, the stability of one of the identified expression sites was confirmed. This site was situated within an intron region of the Cdk6 gene on the CHO-K1 cell chromosome NC_048595.1. Employing CRISPR-Cas9-mediated homology-directed repair, an integration strategy was devised to incorporate the enhanced green fluorescent protein (EGFP) reporter gene into this specific site. Subsequently, the human serum albumin (HSA) gene was cloned using the same method. The study's findings demonstrated that exogenous genes could be stably integrated and expressed at this locus.
The selection of the three ABC sites was informed by high-throughput sequencing results. High-throughput sequencing identified sites with elevated read counts, indicative of gene editing events. We questioned whether these sites could reliably host and express the required antibody genes. This consideration was grounded in our prior experiment, as there was no existing literature for reference. Through high-throughput sequencing, we identified loci with notably high read counts, suggesting potential gene editing events. Consequently, we chose the three loci with the highest read counts for further experimentation.
- A 2011 paper is cited in reference to the use of CRISPR which was only invented in 2013.
Thank you for highlighting this issue. It was, in fact, a mistake in the arrangement of references. We have now corrected it according to the current data.
Grav LM, la Cour Karottki KJ, Lee JS, Kildegaard HF. Application of CRISPR/Cas9 genome editing to improve recombinant protein production in CHO cells. Heterologous Protein Production in CHO Cells: Methods and Protocols. 2017:101-18.
- Re insertion in the Cdk6 gene. Is there an effect on Cdk6 expression?
Thank you for the comment. The insertion of the site does not affect the normal expression of the gene because the inserted site is located in the intron region.
- GF (presumably “green fluorescence” is not defined.
Thank you for bringing this issue in our attention. We have amended.
- Four out of how many?
Thank you for your comment. I would like to inform you that, out of 32, we have selected 4 positive monoclonal clones for further analysis for EGFP. Additionally, for HAS, we currently have a total of 58 monoclonal clones.
- The HSA expression of 16 mg/L – how does this compare to other systems that make HSA? Is this level of expression markedly beKer than other systems?
Thank you for your comment. The HSA expression level of 16 mg/L is a quantitative measure of the amount of HSA produced by a particular expression system. To assess whether this level of expression is considered high or low, you would typically need to compare it to expression levels achieved by other systems or methods. Expression levels can vary depending on the host organism (such as bacteria, yeast, insect cells, or mammalian cells), the specific expression system used (e.g., bacterial expression, yeast expression, mammalian cell expression), and the optimization of various parameters in the production process. In conclusion, 16 mg/L considered a higher expression level, but further evolutions required on large scale level in expression.
- The expression in stable over 60 generations. Is this an effective measure of stability? The authors refer to the need for stability over years, but that would be hundreds of generations. Is there a way to measure stability in comparison to a reference standard?
Thank you for your comment. The expression is stable over 60 generations suggests that the expression of the target protein (EGFP or HSA) remains consistent through multiple generations of the expression system. While this information provides some insight into the stability of expression, it may not be sufficient on its own to determine the overall stability of the system, and it's important to consider additional factors. To assess the stability of protein expression, especially over an extended period of time or multiple generations, it's common to use various analytical methods and compare the results against a reference standard such as quantitative analysis, consistency, genetic stability, cell line stability, post translational modifications and functional assays.
In summary, the statement about stability over 60 generations provides some information, a comprehensive assessment of stability involves a combination of quantitative measurements, genetic analysis, and comparative studies. Comparing results to a reference standard or established benchmarks is a valuable approach to contextualize and validate the further stability of the expression system.
- Figure 5 legend. Include details of statistical treatment in the legend. “P0” to “P60” are not explained in the legend.
Thank you for bringing this issue to our attention. Actually, P stands for the passages (generations) of clones from P0-P60. We have modified the figure according to your suggestions.
Reviewer 2 Report
Comments and Suggestions for Authors
In the present manuscript, the authors propose a particular locus in CHO genome as particularly suitable for CRISPR-Cas-mediated site-specific integration of genes, as this strategy leads to efficient recombinant protein expression. Model proteins used are the intracellularly expressed GFP and the secreted human serum albumin. They show that the cell lines modified in this way maintain the productivity through several (up to 60) passages and can grow at high densities.
The manuscript would strongly benefit from including the description of analytics methods used to measure the secreted HSA (also the required negative controls must be shown), at the moment no data are presented.
Further, the manuscript should be restructured as the Figures do not appear in the correct order and the statements in the Results section are difficult to follow, as the “Materials and Methods” chapter, which appears later, is the one containing the required background information.
The contents of the Discussion Section should be reconsidered, as the authors describe several valuable options to improve their current achieved production yields, but these are not compared with the current state-of-the-art results in similar systems. This information is crucial to the basic argument of the article, namely the functionality of the genomic site suggested for insertion.
Please find below a list of remarks which I hope will be helpful.
Line 23: Keywords: this is HSA, and not HAS
Line 28: „chemical conformation“- maybe composition? Or did you mean medium with few as possible additives?
Line 30: “post-translational stabilization“ – did you mean post-translational modifications?
Figure 3 and b appear in front of 1 and 2, and there is also no mentioning of 1 and 2 in the text.
Figure 3a. The byline described plasmids not mentioned previously – please describe more precisely or modify the Results section.
Figure 3b. The chromatogram has overlapping peaks and the bases in the interpretation do not correspond to the chromatogram peaks – could you please interpret these?
Line 97, and the entire paragraph: Here it is not clear, where the mCherry-transfected cells come from. I think the paper was rewritten and the order of section was modified, nevertheless the text should guide the logical inference of the events. Please see that all information required is provided to the reader in the article as now structured.
Figure 4a. All measured cells should be present in the plots, especially if they are quantified in percent population – please reformat the axis and render the plots new.
Figure 4a: fluorescence, not fluorescent
Line 116: monoclonal cell lines
Line 137: is this deviation that is statistically determined, significant and by chance do you mean 1500u? Please specify.
Figure 6a: fluorescence not fluorescent
Line 175: “Western blot” - which results show this?
Figure 7: I assume that the cell panels labelled “cell viability” pertain to the particular cell line measured – please adjust the labels accordingly.
Line 239: at latest at this point, the results obtained should be compared with what are typical expression levels obtained for comparable systems. Please refer to the literature sources and put at least the HSA-expression results in context with the ones presented. I would also recommend that the yields are accompanied with the data on product amount in pg/cell/day (or another measure of specific productivity of the cells).
Lines 259-264: of course fermentation is a very good perspective option, but these sentences have a lot of the character of forward-looking statements: without an actual reactor experiment or actual attempts to improve the cell host, these are very speculative. I propose exchanging those with an overview what other comparable studies can deliver, in terms of yields and specific productivity.
Lines 274-278: I suggest describing these methods in a perspective of how they can enhance your system, with the described method put into focus.
Line 286: and humidified atmosphere , I assume
Line 293: H2O, 2 in subscript
Line 310: this is HAS, not HAS
Line 314: what is an EP tube?
Line 316: 1% Triton dissolved in which buffer? Protease K activity units, concentration and source are missing
Line 320: please all centrifugation units in g, and add the temperature
Line 320: this is a freezer and not a refrigerator, if is at -30°C
Line 351: results of the Western blot are not presented. Importantly, we miss how the HSA concentration was evaluated.
References 3 and 8 are identical.
References 16 and 29 are identical.
Reference 21: Names and surnames of the authors are not given as requested.
Author Response
Dear Editor
International Journal of Molecular Sciences
SUBJECT: ARTICLE RE-SUBMISSION (REVISED)
The subjected paper entitled “CRISPR-Cas9 mediated stable expression of exogenous proteins in the CHO cell line through site-specific integration” is revised according to the referee’s suggestions. The revision is now up to the mark as advised by the potential reviewers. The revised version of our paper has been submitted for publication.
We hereby present you with a list of all the changes/corrections we made in our revised manuscript, along with the Track Changes feature applied in the main file:
Reviewer-2
We are thankful for the valuable suggestions that have helped enhance the quality of our manuscript. Following our respected reviewer your instructions, we have changed the order of figures and added details in the results from the methodology section to ensure a clear understanding for the readers. We have highlighted changes in yellow, but the figure order and results section modifications are presented in green to address the comments of our first respected reviewer who had the same concern.
Additionally, we have included more details in the discussion section to compare our experimental scheme with previous experiments. We hope our manuscript will contribute to further experimental approaches and provide sufficient information for better understanding
- Line 23: Keywords: this is HSA, and not HAS
Thank you highlighting the issue. It was a typo mistake and now modified.
- Line 28: chemical conformation “- maybe composition?
Thank you highlighting the issue. It was a typo mistake and now modified.
- Line 30: “post-translational stabilization “– did you mean post-translational modifications?
Thank you for bringing the issue in our attention. Modified.
- Figure 3 and b appear in front of 1 and 2, and there is also no mentioning of 1 and 2 in the text.
Thank you for your comment. We have referenced Figure 3a and b in the first section of the results. Furthermore, we have reorganized the order of the figures to align with the manuscript sections.
- Figure 3a. The by-line described plasmids not mentioned previously – please describe more precisely or modify the Results section.
Thank you for the comment. The selection of the three ABC sites was informed by high-throughput sequencing results. High-throughput sequencing identified sites with elevated read counts, indicative of gene editing events. We questioned whether these sites could reliably host and express the required antibody genes. This consideration was grounded in our prior experiment, as there was no existing literature for reference. Through high-throughput sequencing, we identified loci with notably high read counts, suggesting potential gene editing events. Consequently, we chose the three loci with the highest read counts for further experimentation.
- Figure 3b. The chromatogram has overlapped peaks and the bases in the interpretation do not correspond to the chromatogram peaks – could you please interpret these?
Thank you for the comment. We have updated the figure with correct interpretation.
- Line 97, and the entire paragraph: Here it is not clear, where the mCherry-transfected cells come from.
Thank you for the comment. A red fluorescent gene expression cassette, mCherry, was constructed upstream of the 5' homologous arm. Subsequently, the plasmid containing mCherry was transfected into CHO-K1 cells. In accordance with the specified site-specific integration approach, cell lines that underwent successful site-specific integration exhibited expression of the EGFP gene, manifested solely as green fluorescence. Conversely, cell lines experiencing random integration events expressed both the EGFP and mCherry genes, resulting in a dual fluorescence phenotype.
Furthermore, we have restructured the presentation of our findings by altering the order of the results and methodology sections. In this revision, we have integrated certain data from the methodology section into the results section to enhance clarity and comprehension.
- Figure 4a. All measured cells should be present in the plots, especially if they are quantified in percent population – please reformat the axis and render the plots new.
Thank you for bringing the matter to our attention. We have made the necessary modifications as per your instructions.
- Figure 4a: fluorescence, not fluorescent
Thank you for highlighting the issue. We have now made the necessary modifications.
- Line 116: monoclonal cell lines
Thank you for comment. Acknowledged.
- Line 137: is this deviation that is statistically determined, significant and by chance do you mean 1500u? Please specify.
All our results demonstrate statistical significance, as illustrated in the figure accompanied by error bars. Additionally, we have included the average intensity of fluorescence units in the current paragraph.
- Figure 6a: fluorescence not fluorescent
Thank you for drawing attention to the issue. We have now made the necessary modifications.
- Line 175: “Western blot” - which results show this?
Thank you for your suggestion. Unfortunately, during our initial submission, we were unable to include the Western blot figure. We have now addressed this by adding the Western blot assay results for HSA gene expression in CHO-K1 cells.
- Figure 7: I assume that the cell panels labelled “cell viability” pertain to the particular cell line measured – please adjust the labels accordingly.
Thank you for your valuable suggestions. All these figures belong to the HSA as we mentioned in the caption. We have made the necessary modifications to all the figures as per your instructions.
- Line 239: at latest at this point, the results obtained should be compared with what are typical expression levels obtained for comparable systems. Please refer to the literature sources and put at least the HSA-expression results in context with the ones presented. I would also recommend that the yields are accompanied with the data on product amount in pg/cell/day (or another measure of specific productivity of the cells).
Thank you for the suggestions. We have made the necessary modifications to this section and added relevant material to compare the results.
- Lines 259-264: of course, fermentation is a very good perspective option, but these sentences have a lot of the character of forward-looking statements: without an actual reactor experiment or actual attempts to improve the cell host, these are very speculative. I propose exchanging those with an overview what other comparable studies can deliver, in terms of yields and specific productivity.
Thank you for your valuable suggestions. We have compared our results in the above sections, and as per the previous comments, we have already added this section.
- Lines 274-278: I suggest describing these methods in a perspective of how they can enhance your system, with the described method put into focus.
Thank you for your comment. The current system also belongs to the CRISPR system, but unfortunately, we could not find much information. If it's inappropriate to include this section, we can remove it.
- Line 286: and humidified atmosphere, I assume
Thank you for suggestion. Acknowledged.
- Line 293: H2O, 2 in subscript
Thank you for suggestion. Acknowledged.
- Line 310: this is HAS, not HAS
Thank you for suggestion. Acknowledged.
- Line 314: what is an EP tube?
Thank you highlighting the issue. We have changed the exact name Eppendorf tube.
- Line 316: 1% Triton dissolved in which buffer? Protease K activity units, concentration and source are missing
Thank you for the comment. We used Triton X-100 in cell lysis buffers to solubilize membranes and improve the extraction of proteins and other cellular components. The use of protease inhibitors in cell lysis buffers is common to prevent the degradation of proteins during the process of cell lysis. The protease used from Chinese company Magen Biotechnology Co., Ltd. We used 20 mg/mL concentration of Protease K.
- Line 320: please all centrifugation units in g, and add the temperature
Thank you for the suggestion. Acknowledged.
- Line 320: this is a freezer and not a refrigerator, if is at -30°C
Thank you for the suggestion. Amended.
- Line 351: results of the Western blot are not presented. Importantly, we miss how the HSA concentration was evaluated.
Thank you for your comment. We have added the Western blot results and figure. The concentration of HSA was measured using the Western blot along with image analysis software. We developed a standard curve using known concentrations of our protein standard to determine the concentration of HSA in our samples.
- References 3 and 8 are identical.
Thank you for highlighting the issue. We have changed the reference.
- References 16 and 29 are identical.
Thank you for highlighting the issue. We have changed the reference.
- Reference 21: Names and surnames of the authors are not given as requested.
Thank you for highlighting the issue. We have changed the reference.
Round 2
Reviewer 2 Report
Comments and Suggestions for Authors
The authors have introduced the corrections and they also show the important fact that their cell development method leads to efficient secretion of one of the relevant model proteins studied (HSA). Unfortunately, in this experiment no negative control (untransformed or mock-transformed strain expression) is included. They could also kindly comment on the yields achieved by other similar systems, as mentioned in the first round of review.
I am sorry that something is wrong with the order of figures: some (such as Nr.2) are not there, and in the text sometimes they appear spelled as Fig. or Figure, which makes the intentions of authors regarding the display elements difficult to follow.
Line 2: Title: I propose the change the title by adding: “supportive evidence for a favorable insertion site at the C12orf35”, or “ at a novel site within…” , or similar – all intended to highlight the actual achievements of your research contribution
Line 146. Figure 3 follows Figure 1 without Figure 2 in between.
Line 208: Legends do not fit the Figure, there are display elements a and b that appear twice
Line 231: 106, 6 in superscript
Line 395: Thank you for undertaking the required correction. Nevertheless, “dependable” is an expression much more pertaining to human resources, promising that someone will fulfil the task by the best of their ability – each and every time. I suppose “reliable”, or similar, would better suit your arguments.
Line 439: “at humified environment 37°C with 5% CO2” should be changed to in humidified environment at 37°C with 5% CO2, 2 in subscript
Line 494: Antibodies should be cited with RRIDs (or otherwise catalogue numbers, if RRIDs are not there)
Author Response
Dear Editor
International Journal of Molecular Sciences
SUBJECT: ARTICLE RE-SUBMISSION (REVISED)
The subjected paper entitled “CRISPR-Cas9 mediated stable expression of exogenous proteins in the CHO cell line through site-specific integration” is revised according to the referee’s suggestions. The revision is now up to the mark as advised by the potential reviewers. The revised version of our paper has been submitted for publication.
We hereby present you with a list of all the changes/corrections we made in our revised manuscript (purple color), along with the Track Changes feature applied in the main file:
Reviewer-2
Dear respected reviewer, we extend our sincere gratitude for your valuable feedback during the first revision, which has significantly enhanced the quality of our manuscript. Your insightful suggestions have prompted further improvements, and we are optimistic that this revised version will contribute significantly to the specific area of research.
- The authors have introduced the corrections and they also show the important fact that their cell development method leads to efficient secretion of one of the relevant model proteins studied (HSA). Unfortunately, in this experiment no negative control (untransformed or mock-transformed strain expression) is included. They could also kindly comment on the yields achieved by other similar systems, as mentioned in the first round of review.
I am sorry that something is wrong with the order of figures: some (such as Nr.2) are not there, and in the text sometimes they appear spelled as Fig. or Figure, which makes the intentions of authors regarding the display elements difficult to follow.
Thank you for the comment. In response to your comments, we would like to address a specific aspect of our study. We conducted a comparison by culturing CHO cells (blank cells) without the HSA gene for 60 generations simultaneously. Remarkably, we observed that their cell viability and cell density closely mirrored those of CHO cells transfected with the HSA gene. However, it is noteworthy that the HSA content in the blank cells was undetectable.
As a result, we made the decision not to include the control group with zero HSA content in the graph. This choice aligns with the methodology employed in similar studies, including the referenced articles, where the HSA content of the control group was also omitted from the graph due to its consistent absence (zero). For your convenience, we have included the cell density and cell viability diagram of the control cells (blank cells) below for your review.
Further, we also modified the figures as per your suggestions. Now, all the figures style is uniform.
Ding, X., Chen, Y., Yang, Z. et al. A comprehensive evaluation of stable expression “hot spot” in the ScltI gene of Chinese hamster ovary cells. Appl Microbiol Biotechnol 107, 1299–1309 (2023).
Ahmadi, MaryamMahboudi, FereidounAhmadi, SamiraEbadat, SaeedehNematpour, FatemehEidgahi, Mohammad Reza AkbariDavami, Fatemeh.PhiC31 integrase can improve the efficiency of different construct designs for monoclonal antibody expression in CHO cells[J]. Protein Expression and Purification, 2017, 134.
Zhou S , Chen Y , Gong X ,et al.Site-specific integration of light chain and heavy chain genes of antibody into CHO-K1 stable hot spot and detection of antibody and fusion protein expression level[J].Preparative biochemistry & biotechnology, 2019, 49(4):384-390.
- Line 2: Title: I propose the change the title by adding: “supportive evidence for a favourable insertion site at the C12orf35”, or “at a novel site within…”, or similar – all intended to highlight the actual achievements of your research contribution
Thank you for your valuable suggestion. I have a kind request: after changing the title, it does not appear as graceful, and upon reviewing various articles, we noticed that specific sites are not mentioned in titles. Kindly consider this perspective. However, we have explicitly stated the site in both the abstract and conclusion sections. If you still believe that it's necessary to alter the title, we are open to making the change. Thank you.
- Line 146. Figure 3 follows Figure 1 without Figure 2 in between.
Thank you for bringing this to our attention. Following your suggestions, we have rearranged the order of the figures. Occasionally, when opening the file on a different laptop, we noticed that the figure order could change. I have personally observed this situation as well. However, we have now made further modifications to address this issue.
- Line 208: Legends do not fit the Figure, there are display elements a and b that appear twice
Thank you for bringing this issue to our attention. We have made the necessary changes.
- Line 231: 106, 6 in superscript
Thank you for the comment. Acknowledged.
- Line 395: Thank you for undertaking the required correction. Nevertheless, “dependable” is an expression much more pertaining to human resources, promising that someone will fulfil the task by the best of their ability – each and every time. I suppose “reliable”, or similar, would better suit your arguments.
Thank you for the comment. Amended.
- Line 439: “at humified environment 37°C with 5% CO2” should be changed to in humidified environment at 37°C with 5% CO2, 2 in subscript
Thank you for the comment. We have replaced the sentence.
- Line 494: Antibodies should be cited with RRIDs (or otherwise catalogue numbers, if RRIDs are not there)
Thank you for the comment. Amended.
Other modifications: We revised the complete manuscript and improved the grammatical errors.
We hope that the corrections, changes, and additions are satisfactory and meet the publication standards of the journal. We look forward to hearing from you.
Best regards
X.M Ding and Colleagues
